# The simplest construction of single-site catalysts by the synergism of micropore trapping and nitrogen anchoring

Zhiqi Zhang [1], Yugang Chen[1], Liqi Zhou[2], Chi Chen[3], Zhen Han[2], Bingsen Zhang [4], Qiang Wu[1], Lijun Yang[1], Lingyu Du[1], Yongfeng Bu[1], Peng Wang [2,5], Xizhang Wang[1], Hui Yang[3] & Zheng Hu [1]

Single-site catalysts feature high catalytic activity but their facile construction and durable utilization are highly challenging. Herein, we report a simple impregnation-adsorption method to construct platinum single-site catalysts by synergic micropore trapping and nitrogen anchoring on hierarchical nitrogen-doped carbon nanocages. The optimal catalyst exhibits a record-high electrocatalytic hydrogen evolution performance with low overpotential, high mass activity and long stability, much superior to the platinum-based catalysts to date. Theoretical simulations and experiments reveal that the micropores with edge-nitrogen-dopants favor the formation of isolated platinum atoms by the micropore trapping and nitrogen anchoring of $[PtCl_6]^{2-}$, followed by the spontaneous dechlorination. The platinum-nitrogen bonds are more stable than the platinum-carbon ones in the presence of adsorbed hydrogen atoms, leading to the superior hydrogen evolution stability of platinum single-atoms on nitrogen-doped carbon. This method has been successfully applied to construct the single-site catalysts of other precious metals such as palladium, gold and iridium.

[1] Key Laboratory of Mesoscopic Chemistry of MOE and Jiangsu Provincial Lab for Nanotechnology, School of Chemistry and Chemical Engineering, Nanjing University, 210023 Nanjing, China. [2] National Laboratory of Solid State Microstructures, College of Engineering and Applied Sciences, and Collaborative Innovation Center of Advanced Microstructures, Nanjing University, 210093 Nanjing, China. [3] Shanghai Advanced Research Institute, Chinese Academy of Sciences, 201210 Shanghai, China. [4] Shenyang National Laboratory for Materials Science, Institute of Metal Research, Chinese Academy of Sciences, 110016 Shenyang, China. [5] Research Center for Environmental Nanotechnology (ReCENT), Nanjing University, 210023 Nanjing, China. These authors contributed equally: Zhiqi Zhang, Yugang Chen, Liqi Zhou. Correspondence and requests for materials should be addressed to Q.W. (email: wqchem@nju.edu.cn) or to P.W. (email: wangpeng@nju.edu.cn) or to Z.H. (email: zhenghu@nju.edu.cn)

Single-site catalysts (SSCs), emerging as a new class of catalysts, feature the extraordinary catalytic activity and selectivity owing to the unsaturated coordination configuration of active centers, quantum size effect and support effect, which bridges the gap between the homogeneous and heterogeneous catalysis[1–4]. The SSCs of precious metals are particularly attractive, which represents the utmost utilization of precious metals. To date, the SSCs have exhibited excellent catalytic performance in some important reactions such as oxidation/hydrogenation reaction, water-gas shift and electrocatalysis[3,5]. Most synthetic methods involve the special equipment, or complicated operation, or expensive precursors[6–8], and the SSCs usually tend to aggregation during the catalytic process[9]. Hence, the facile construction and durable use of SSCs is a challenging and wide interesting topic today.

As is known, the formation and catalytic durability of SSCs are heavily associated with the metal-support interactions, which is a basic consideration in SSCs studies. Actually, most SSCs are obtained by enhancing metal-support interactions, e.g., by introducing dopants or defects to the supports or confining the single atoms inside the porous supports[10]. In recent years, we reported the 3D hierarchical carbon-based nanocages, which are characterized by the network geometry, coexisting micro-meso-macropore structure, high specific surface area, and convenience for nitrogen-doping[11–13]. These characteristics, especially the natural coupling of micropores (~0.6 nm) and N-dopant, are the unique advantage to reinforce the interaction with single metal atoms by combining the physical confinement of micropores and the chemical bonding of nitrogen dopant. With this in mind, herein the single atomic Pt has been immobilized onto the hierarchical nitrogen-doped carbon nanocages (hNCNC) by a simple impregnation-adsorption method just via a solution adsorption of anions followed by a mild drying below 70 °C. The optimal $Pt_1$/hNCNC SSC exhibits a record-high electrocatalytic performance for the hydrogen evolution reaction (HER) in acidic medium with a low overpotential, high mass activity and long stability, much better than the Pt-based catalysts to date including Pt/C benchmark. The formation mechanism and superior stability of $Pt_1$/hNCNC are understood by density functional theory (DFT) calculation. This strategy has been successfully extended to construct the SSCs of other precious metals including Pd, Au and Ir. This finding provides a generalized simple method to construct durable SSCs of precious metals by combining micropore trapping and dopant anchoring effects with doped-carbon supports full of micropores.

## Results

**Structure characterization and formation mechanism.** The metal-support interaction is necessary for the stabilization of singly dispersed metal atoms[10]. As revealed by our previous DFT calculation, transition metal atoms show enhanced binding energies with N-doped $sp^2$ carbon in comparison with undoped carbon supports owing to the nitrogen participation[14]. The hNCNC support used in this study is a typical one with coexisting micro-meso-macropore structure, a high nitrogen content of 9.5 at.%, and a specific surface area of 877 $m^2 g^{-1}$. For comparison, the hierarchical carbon nanocages (hCNC) support with similar morphologic and structural characteristics but without nitrogen doping is also used for loading Pt. Worthy to emphasize is that both hNCNC and hCNC possess abundant intrinsic micropores across the carbon shell with the size of ~0.6 nm[12], which favors the physical confinement of metal atoms (Supplementary Fig. 1).

By the simplest impregnation-adsorption followed by filtration and washing, Pt/hNCNC and Pt/hCNC catalysts were prepared with the high Pt loading of 2.92 wt% (see Methods and Supplementary Fig. 2). Figure 1 displays the morphological and structural characterizations of the catalysts. The high-angle annular dark-field (HAADF) scanning transmission electron microscopy (STEM) image of Pt/hNCNC clearly shows that the individual Pt atoms randomly disperse on the hNCNC support, which account for a high percentage of 96.7% in all the Pt dots (Fig. 1a, Supplementary Figs. 3 and 4). By X-ray absorption fine structure (XAFS) analysis, Pt/hNCNC exhibits stronger white-line intensity than the control samples of Pt foil and hNCNC-supported Pt nanoparticles in the normalized X-ray absorption near-edge structure (XANES) spectra (Fig. 1c), suggesting the partial oxidation state of Pt atoms in Pt/hNCNC relative to Pt foil and nanoparticles[15]. As further revealed by the $k^3$-weighted extended XAFS (EXAFS) at the Pt $L3$-edge, Pt/hNCNC has a dominant peak at 1.75 Å from Pt-N/C/O with the absence of peak at 2.61 Å from Pt-Pt contribution (Fig. 1d, Supplementary Fig. 5 and Table 1). This result confirms the isolated state of Pt single atoms, in consistent with the HAADF-STEM observation (Fig. 1a). Hence, the Pt/hNCNC catalyst is indeed the SSC, renamed as $Pt_1$/hNCNC hereafter.

For Pt/hCNC, most Pt atoms exist in the isolated state, but a slight aggregation could be observed (Fig. 1b, Supplementary Fig. 4). The white-line of Pt/hCNC is a little more intensive than that of $Pt_1$/hNCNC, indicating a slightly higher oxidation state of Pt atoms in Pt/hCNC than in $Pt_1$/hNCNC (Fig. 1c). This result also indicates the electron transfer from Pt to hNCNC is slightly less than that to hCNC due to the N participation. In addition to the dominant peak at 1.89 Å, a trace peak at ~2.55 Å indicates the existence of a little Pt-Pt coordination (coordination number $n = 0.4$) from Pt clusters (Fig. 1d, Supplementary Fig. 5 and Table 1), in agreement with the HAADF-STEM image (Fig. 1b). These results indicate the micropores alone have the capability for trapping Pt single atoms as the case of Pt/hCNC, and the synergic micropore trapping and nitrogen anchoring much strengthen the formation capability of Pt single atoms as the case of $Pt_1$/hNCNC.

Pt 4f XPS spectra can be deconvolved into two sets of peaks for Pt 4$f_{7/2}$ and Pt 4$f_{5/2}$ (Fig. 1e). The binding energies of Pt 4f are 72.4 and 75.7 eV for $Pt_1$/hNCNC, and 72.7 and 76.0 eV for Pt/hCNC. In comparison with the Pt nanoparticles supported in hNCNC (denoted as Pt-NPs/hNCNC) with the binding energies of 71.4 and 74.7 eV, Pt atoms in $Pt_1$/hNCNC and Pt/hCNC are partially oxidized, in agreement with the XANES results (Supplementary Fig. 6). In addition, the lower binding energy of $Pt_1$/hNCNC than Pt/hCNC indicate the higher electron density of Pt atoms in $Pt_1$/hNCNC. Accordingly, the XPS signal for pyridinic N (py-N) in $Pt_1$/hNCNC shows a slight widening to the higher binding energy side in comparison with that in pristine hNCNC support (Fig. 1f, Supplementary Fig. 7), which is attributed to the electron transfer rewarded from py-N to the anchored Pt single atom. The results indicate that the $Pt_1$-hNCNC interaction is stronger than the Pt-hCNC interaction due to the nitrogen participation for the former, in agreement with the EXAFS results (Fig. 1d). In other words, the nitrogen dopants are helpful for anchoring single Pt atoms and preventing the diffusion/aggregation.

The formation of Pt single atoms on hNCNC and hCNC was understood with DFT modeling. More than ten configurations were adopted to check the capability of micropores to capture $[PtCl_6]^{2-}$ anions. Herein, six typical configurations and corresponding adsorption free energies are presented in Fig. 2. The calculation results indicate their capability to capture $[PtCl_6]^{2-}$ anions due to the exothermic adsorption free energy, especially for the micropores in bi-layer models (4, 5, 6). Since hNCNC and hCNC possess the abundant intrinsic defective micropores with the size of ~0.6 nm across the carbon shell of ca. 2–5 graphitic

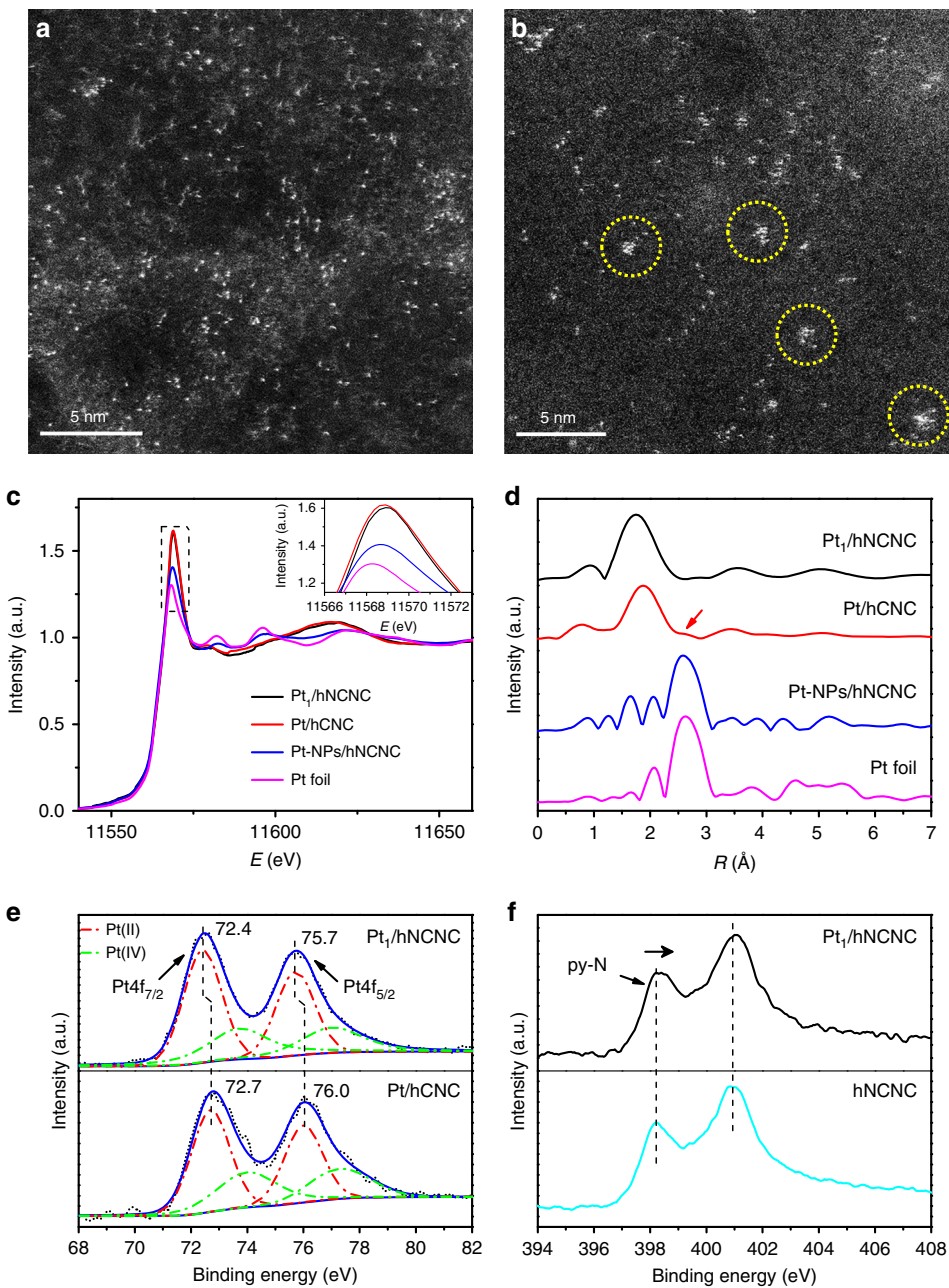

**Fig. 1** Morphological and structural characterizations of Pt₁/hNCNC and Pt/hCNC. **a**, **b** HAADF-STEM images of Pt₁/hNCNC and Pt/hCNC, respectively. The circles in **b** mark the slight aggregation of Pt atoms. **c** Normalized XANES spectra at the Pt $L_3$ edge. Inset is the local enlargement. **d** $k^3$-weighted R-space Fourier transformed spectra from EXAFS. **e** XPS spectra for Pt 4f. **f** XPS spectra for N 1s. In (**c**, **d**, and **f**), the corresponding data for Pt-NPs/hNCNC, Pt foil and hNCNC are presented for comparison

layers, models (4, 5, 6) are more close to the practical situation. For hCNC, the $[PtCl_6]^{2-}$ is trapped in the micropore with an adsorption free energy of 4.3 eV, much larger than 2.8 eV for the case of graphitic plane. Hence, the $[PtCl_6]^{2-}$ anions are easily captured on hCNC by micropore trapping, leading to the formation of most Pt single atoms with only a slight aggregation (Fig. 1b). The micropore trapping mainly comes from the van der Waals interaction between the $[PtCl_6]^{2-}$ anion and the atoms at the edge (or wall) of micropore. The trapping strength is closely related to the effective atoms involved in the van der Waals interaction. Accordingly, the trapping of $[PtCl_6]^{2-}$ in the bi-layer micropore (4.3 eV) is much stronger than the case in the mono-layer micropore (2.3 eV). The matchable size of micropore (~0.6 nm) to $[PtCl_6]^{2-}$ (~0.5 nm) also contributes a lot to the strong

capturing capability. With N participation, the adsorption free energy is further increased to 4.6 or 4.9 eV for models 5 and 6, respectively, indicating the enhanced capability to stabilize $[PtCl_6]^{2-}$ anions due to the synergic micropore trapping and nitrogen anchoring. Specifically, the py-N atoms at the edges of the micropores are protonated and positively charged in the $H_2PtCl_6$ solution[16], leading to the formation of stable ion pair of $[C_x(NH)_2]^{2+}[PtCl_6]^{2-}$ via the electrostatic interaction. As for the graphitic N-doped graphene sheet, the adsorption free energy is quite similar to that of the pristine one as revealed by DFT calculation, indicating the little contribution to the capture of $[PtCl_6]^{2-}$ (Supplementary Fig. 8). The micropore trapping and the electrostatic interaction prevent the $[PtCl_6]^{2-}$ anions escaping from the micropores.

The $[PtCl_6]^{2-}$ anions trapped in the micropores are subsequently dechlorinated upon heat treatment at 70 °C as confirmed by mass spectroscopy analysis, leaving the trapped Pt single atoms (Supplementary Fig. 9). The trapped Pt single atom possesses the adsorption energy of 1.81, 1.92, 2.21, and 2.34 eV for models 1, 4, 5, 6, respectively, which indicates the more stable existence of Pt single atoms trapped in hNCNC than in hCNC and on graphene sheet (Supplementary Fig. 10). Based on the preceding experimental and theoretical results, the synergism of micropore trapping and nitrogen anchoring is the most favorable for trapping the $[PtCl_6]^{2-}$ anions and the derived Pt single atoms, leading to the highly dispersive Pt single atoms on hNCNC. In contrast, without the nitrogen anchoring, the sole micropore trapping presents a little inferior interaction with the $[PtCl_6]^{2-}$ anions and the derived Pt single atoms, leading to the slight aggregation of the Pt atoms in Pt/hCNC in comparison with the case in $Pt_1$/hNCNC. To our knowledge, this is the first example to construct SSCs by taking advantage of the synergism of micropore trapping and N-dopant anchoring, which could provide an advanced route for convenient construction of various SSCs.

**HER electrocatalytic activity**. As is known, the HER via water electrolysis is an important route for the production of clean and sustainable hydrogen, in which the precious Pt-based electrocatalysts are the most effective[17]. We have evaluated the HER performance of the $Pt_1$/hNCNC in comparison with the cases of different Pt catalysts, as shown in Fig. 3. It is worth mentioning that a graphite rod rather than a Pt wire was used as the counter electrode to avoid the controversy over the origin of the HER activity[18]. A series of $Pt_1$/hNCNC catalysts with different Pt loading (0.75~5.68 wt%) were prepared for optimizing the HER performance (Supplementary Fig. 2). The overpotentials (vs. RHE) at the current density of 10 mA cm$^{-2}$ decrease from 40 to 20, 15, and 13 mV with increasing Pt loading from 0.75 to 1.48, 2.92, and 5.68 wt%, respectively, demonstrating a convergence to the lowest overpotential (Fig. 3a, c). For the precious metal catalysts, mass activity is also an important parameter. At the overpotential of 20 mV, the mass activities for the $Pt_1$/hNCNC-0.75, -1.48, -2.92, and -5.68 catalysts are 5.10, 7.01, 7.60, and 5.75 A mg$^{-1}$ $_{Pt}$, respectively, showing a volcano-type change with the maximum for the $Pt_1$/hNCNC-2.92 (Fig. 3c). Despite the slight (only 2 mV) lower overpotential for $Pt_1$/hNCNC-5.68 than $Pt_1$/hNCNC-2.92, the mass activity for the former is much (ca. 24 %) smaller than the latter (5.75 vs. 7.60 A mg$^{-1}$ $_{Pt}$) due to the slight aggregation of Pt atoms (Supplementary Fig. 11). Based on the combination of low overpotential and high mass activity, the $Pt_1$/hNCNC-2.92 catalyst presents the best HER performance. In addition, the overpotential (15 mV) and Tafel slope (24 mV dec$^{-1}$) of $Pt_1$/hNCNC-2.92 catalyst is lower than the corresponding ones of the Pt/hCNC catalyst (17 and 28 mV dec$^{-1}$) of the same loading (Fig. 3a, b). The slight inferior HER performance of the Pt/hCNC to the $Pt_1$/hNCNC should mainly result from the slight aggregation for the former (Fig. 1a, b). It is worth noting that the mass activity of the optimal $Pt_1$/hNCNC catalyst (7.60 A mg$^{-1}$ $_{Pt}$) is about 18.5-fold higher than that of the 20 wt% commercial Pt/C (0.41 A mg$^{-1}$ $_{Pt}$) at the overpotential of 20 mV, indicating the much enhanced Pt utilization in the HER. In addition, the turnover frequency (TOF) at the overpotential of 20 mV is 7.67 s$^{-1}$ for $Pt_1$/hNCNC, much higher than that of the commercial Pt/C catalyst (0.41 s$^{-1}$), showing the superb electrocatalytic activity for the former[19].

The preceding results indicate that the $Pt_1$/hNCNC catalyst shows the much lower overpotential and Tafel slope, and the much higher mass activity and TOF value in the HER than the corresponding ones of the commercial Pt/C catalysts. Actually, the HER performance of $Pt_1$/hNCNC is superior to the most reported catalysts to date (Supplementary Table 2). The excellent HER performance should result from the charge transfer from Pt to the support as reflected by the high oxidation state of Pt atoms

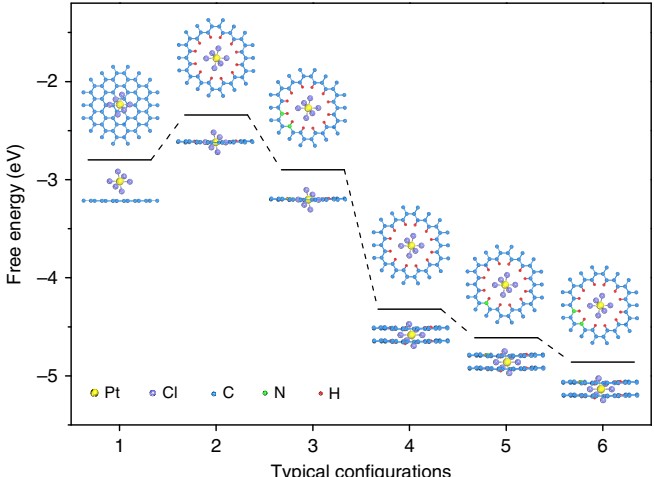

**Fig. 2** Six typical configurations of $[PtCl_6]^{2-}$ on different supports and corresponding calculated free energies. **1** Graphene sheet. **2** Graphitic mono-layer with a micropore of 0.6 nm. **3** Graphitic mono-layer with the micropore decorated by two py-N atoms. **4** Graphitic bi-layer with a micropore of 0.6 nm. **5** Graphitic bi-layer with the micropore decorated by one py-N atom. **6** Graphitic bi-layer with the micropore decorated by two py-N atoms

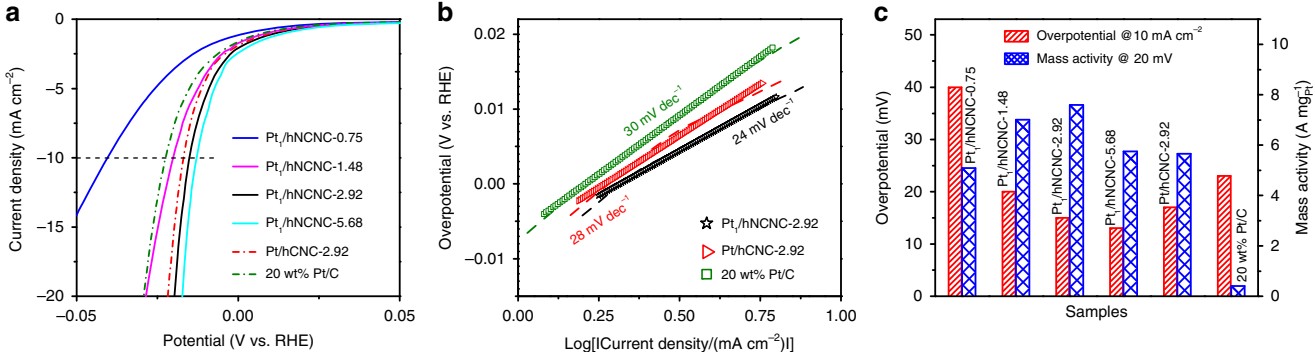

**Fig. 3** HER performance of $Pt_1$/hNCNC in 0.5 mol L$^{-1}$ $H_2SO_4$ solution at a scan rate of 5 mV s$^{-1}$ after *iR*-compensation. **a** Polarization curves. **b** Tafel plots. **c** Overpotentials at 10 mA cm$^{-2}$ and mass activities at 20 mV (vs. RHE) of the series of catalysts. The data for Pt/hCNC and commercial Pt/C (20 wt% Pt) are presented for comparison

in Pt$_1$/hNCNC (Fig. 1c), which leads to the unoccupied 5d orbitals of Pt single atoms and thus favors the HER[20].

**Stability of Pt$_1$/hNCNC catalysts.** The evolutions of the polarization curves and morphologies of the Pt$_1$/hNCNC, Pt/hCNC and commercial Pt/C were obtained before and after 5000 and 10000 CV scans to evaluate their HER stability, in combination with the DFT simulations, as presented in Fig. 4. The overpotential at the current density of 50 mA cm$^{-2}$ for Pt$_1$/hNCNC only slightly increased from the initial 29.5 mV through 30.9 mV after 5000 cycles to 31.7 mV after 10000 cycles. In contrast, before and after 5000 and 10000 CV scans, the corresponding overpotentials are 31, 39.5, and 47.4 mV for Pt/hCNC, and are 41.4, 48.7, and 56.8 mV for commercial Pt/C, showing much faster deterioration rate than that for Pt$_1$/hNCNC (Fig. 4a). The HAADF-STEM images indicate that, after 10000 CV scans, the Pt$_1$/hNCNC keeps its characteristic of the single-atom dispersion while the Pt/hCNC presents an obvious aggregation into clusters or even Pt nanoparticles in comparison with the original ones (Fig. 1a, b, Fig. 4b, c). These results demonstrate the extraordinary stability of the Pt$_1$/hNCNC much superior to those of the Pt/hCNC and the commercial Pt/C, and also indicate the direct correlation of the HER-performance deterioration with the Pt aggregation.

The preceding results indicate that Pt$_1$/hNCNC possesses superb electrocatalytic activity and stability for the HER, which is associated with the capability of hNCNC to prevent Pt atoms from diffusing and aggregating. To understand this point, two typical configurations, i.e., Pt bonding with two py-N atoms (PtN2) or two carbon atoms (PtC2) to model Pt$_1$/hNCNC or Pt/hCNC respectively, were used to simulate the hydrogen evolution and the competing substrate hydrogenation coexisting in the HER process for comparison (Fig. 4d, e). The substrate hydrogenation involves the H insertion in between the Pt and N for Pt$_1$/hNCNC or the Pt and C for Pt/hCNC, which will lead to the detachment of Pt atoms from the substrate and subsequent diffusion and aggregation. For the PtN2, the hydrogen evolution pathway is preferred to the substrate hydrogenation due to the much larger drop of free energy for the former (Fig. 4d). In contrast, for the PtC2, the free energy profiles of the two pathways are entangled in each other, indicating the parallel occurrence of the two reactions (Fig. 4e). Hence, during the HER process, the competing substrate hydrogenation is much weaker for Pt$_1$/hNCNC than Pt/hCNC, leading to the much better stability for the former.

Thanks to the synergic micropore trapping and nitrogen anchoring, the Pt SSCs present excellent resistance to the diffusion and aggregation upon heating. The Pt$_1$/hNCNC catalyst remains the high HER activity after additional heat treatment at 150 °C for 1 h in Ar. The high thermal stability of the Pt$_1$/hNCNC suggests its applicability to most electrocatalytic reactions. With a heat-treatment at 225 and 300 °C for 1 h, the HER activity deteriorates remarkably with an overpotential increase from the initial 15 to 25, and 84 mV at 10 mA cm$^{-2}$, respectively, due to the aggregation of Pt atoms (Supplementary Fig. 12).

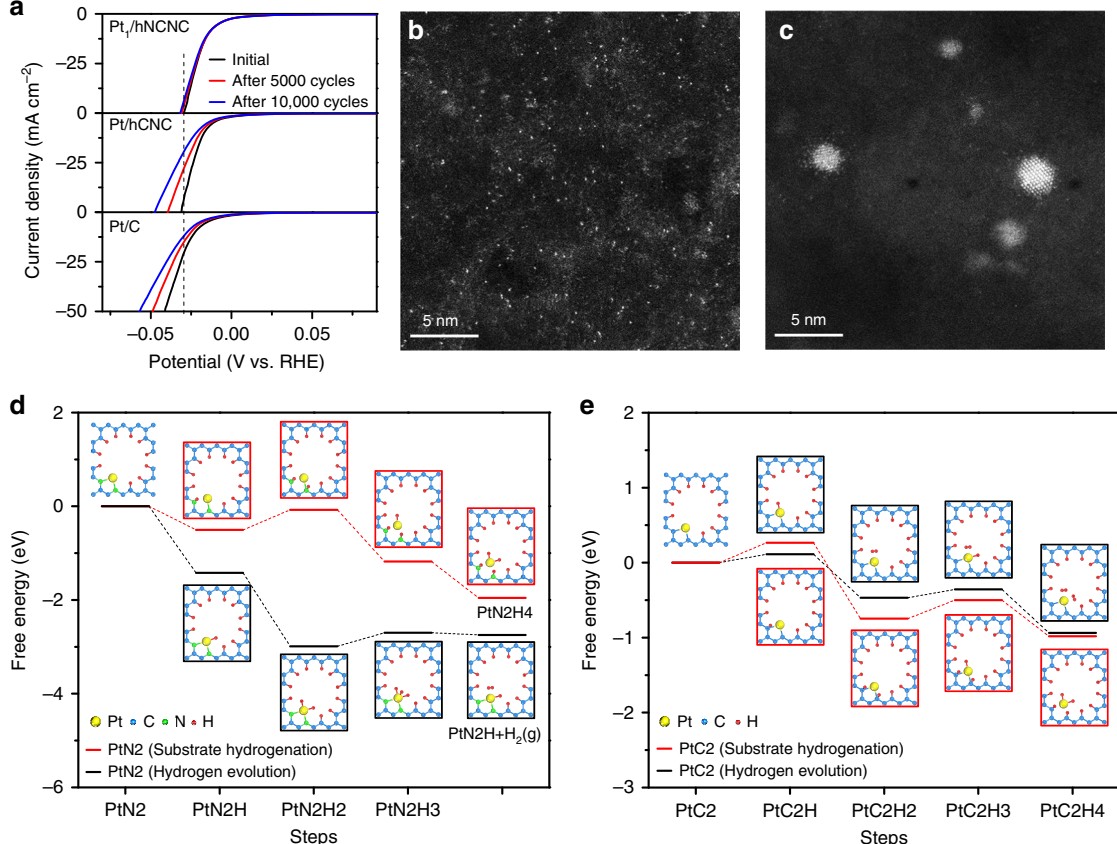

**Fig. 4** HER durability of Pt$_1$/hNCNC, Pt/hCNC and commercial Pt/C in 0.5 mol L$^{-1}$ H$_2$SO$_4$. **a** Polarization curves before and after 5000 and 10000 CV scans between 0 and −0.4 V (vs. Ag/AgCl). **b, c** HAADF-STEM images of Pt$_1$/hNCNC (**b**) and Pt/hCNC (**c**) after 10000 cycles of CV scan. **d, e** Free energy of the hydrogen evolution and substrate hydrogenation for a Pt atom bonding with two py-N atoms (PtN2) or two carbon atoms (PtC2), respectively. The corresponding optimized structures are presented there

## Discussion

As is known, carbon-based supports are the typical ones employed for constructing SSCs. Usually, the single-atom dispersion is achieved through complex procedures or harsh conditions, such as the high-temperature treatment[15,21–24], iced photochemical reduction[8], and atomic layer deposition[20,25]. Herein, we have developed the super simple construction method just via a solution adsorption of anions followed by a mild drying below 70 °C, by making use of the synergism of micropore trapping and nitrogen anchoring of the carbon support. To our knowledge, this method is the simplest one among all the preparation methods of SSCs with different supports. The as-constructed $Pt_1$/hNCNC catalyst possesses superb electrocatalytic hydrogen evolution performance with low overpotential, high TOF, high mass activity and long durability (Fig. 3, Fig. 4a, b), and could be highly promising for other catalytic reactions owing to its unique characteristics. Moreover, this method is also applicable to prepare the SSCs of other precious metals such as Pd, Au and Ir as shown in Fig. 5 (Supplementary Fig. 13). Their chemical components were confirmed with the energy-dispersive X-ray spectroscopy (EDS) elemental mapping (Supplementary Fig. 14). The convenient construction and generalized applicability of this strategy is indeed significant for promoting the fundamental study of SSCs as well as their practical applications.

In summary, we have developed the simplest impregnation-adsorption method to prepare the single-site Pt catalysts just via a solution adsorption of anions followed by a mild drying below 70 °C by making use of the synergism of micropore trapping and nitrogen anchoring of carbon support. The optimal $Pt_1$/hNCNC shows superb activity and stability toward the HER, much superior to the commercial Pt/C benchmark, locating at the top level for HER catalysts to date. The combined experimental and theoretical studies reveal that the synergism of micropore trapping and nitrogen anchoring is the most favorable for trapping the $[PtCl_6]^{2-}$ anions and the derived Pt single atoms, leading to the convenient formation and high stability of the Pt SSCs with a high loading. This method is also applicable to prepare the SSCs of other precious metals such as Pd, Au, and Ir. The convenient construction and generalized applicability of this strategy is significant for promoting the fundamental study of SSCs as well as their practical applications.

## Methods

**Materials**. hNCNC and hCNC supports were synthesized at 800 °C by the in situ MgO template method with pyridine and benzene precursors[11–13], respectively. Commercial Pt/C (20 wt% Pt) and $H_2PtCl_6 \cdot 6H_2O$ were purchased from Shanghai Hesen Electric Co., Ltd. and Sinopharm Chemical Reagent Co., Ltd. respectively.

**Synthesis of $Pt_1$/hNCNC and the control catalysts**. The $Pt_1$/hNCNC catalysts were prepared by a simple impregnation-adsorption method. Briefly, 50 mg hNCNC was dispersed in 80 mL distilled water, and then an appropriate amount of $H_2PtCl_6 \cdot 6H_2O$ solution (2 mg mL$^{-1}$) was added dropwise, followed by stirring at 70 °C for 10 h. The product was obtained by filtration, repeated washing with water

and ethanol, and drying at 70 °C overnight. A series of Pt/hNCNC catalysts were synthesized with different feeding amount of $H_2PtCl_6 \cdot 6H_2O$ solution of 0.5, 1, 2, and 4 mL. After getting the optimized Pt loading, the control catalyst of Pt/hCNC was prepared similarly by using undoped hCNC for comparison. Pt-NPs/hNCNC catalyst was synthesized by a microwave-assisted ethylene glycol reduction method as described in our recent study[26].

**Catalyst characterization**. For transmission electron microscopy characterization, the catalysts were dispersed in ethanol via ultrasonic treatment, and drop cast onto holey carbon films. The atomic resolution HAADF images were acquired on an aberration-corrected STEM Titan[3] cubed with a field emission gun at 300 kV. The probe convergence semi-angle was 20.5 mrad and the angular range of the HAADF detector was from 60.5 to 200 mrad. The beam current of 41 pA was used with corresponding dose of $1.15 \times 10^4$ e/Å$^2$. A 2D N-$K$ edge elemental mapping was acquired using STEM electron energy loss spectroscopy (EELS) spectrum imaging on a Gatan Quantum 966 system. The dwell time for each spectrum were typically 0.5 s with a collection-angle of 35.6 mrad for the core-loss spectra (300–825 eV). To increase signal and noise ratio, the final spectrum of N-$K$ edge was summed over entire elemental mapping area. The background was subtracted using a power law fitting. STEM-EDS elemental mappings of Pt, Pd, Au, and Ir were carried out on a FEI Tecnai F20 TEM at 200 kV.

X-ray photoelectron spectroscopy (XPS) was measured on VG ESCALAB MKII. The XAFS at the Pt $L_3$-edge was obtained on BL14W1 beamline in the Shanghai Synchrotron Radiation Facility (SSRF), Shanghai Institute of Applied Physics, China, operated at 3.5 GeV with injection currents of 140–210 mA. In the measurement, a Si(111) double-crystal monochromator was used to reduce the harmonic component of the monochrome beam. Pt foil were used as reference samples and measured in the transmission mode. $Pt_1$/hNCNC, Pt/hCNC and Pt-NPs/hNCNC were measured in fluorescence mode. The specific surface area and pore structure of hNCNC and hCNC were measured on a Thermo Fisher Scientific Surfer Gas Adsorption Porosimeter at 77 K. The Pt content in the catalyst was measured by ultraviolet visible spectroscopy (UV-vis, Shimadzu UV-3600). Briefly, an appropriate volume of $H_2PtCl_6 \cdot 6H_2O$ solution (2 mg mL$^{-1}$) was added to deionized water and diluted to 100 mL. After the impregnation-adsorption by hNCNC, the residual solution was measured by UV-vis, and by the comparison of the absorbance with the working curve, the Pt content in hNCNC was deduced.

**Electrochemical characterization**. Electrochemical measurements were performed on a CHI 760E workstation (CH Instruments) using a three electrode system at 25 °C in 0.5 mol L$^{-1}$ $H_2SO_4$ (degassed with $N_2$). A graphite rod and a Ag/AgCl (3 M KCl) electrode were used as the counter and reference electrode, respectively. The working electrode was fabricated as follows: 2 mg of catalyst was dispersed ultrasonically in a mixed solution of 800 μL of water, 200 μL of ethanol, and 40 μL of Nafion (Dupont, 5 wt%). Then 10 μL of fresh catalyst ink was dropped onto a glassy-carbon electrode (GC; 5 mm in diameter, Pine Instrument Co.), and dried at room temperature for 12 h. During the test, the working electrode was rotated at 1600 rpm for removing $H_2$ bubbles. All the potentials measured were converted to the reversible hydrogen electrode (RHE), i.e., $E_{RHE} = E_{Ag/AgCl} + 0.209 + 0.059 \times pH$. All the electrochemical measurements are represented with $iR$-compensation by using the CHI software.

**DFT calculations**. The spin-unrestricted DFT calculations were carried out by the DMol3 code from Accelrys[27,28]. The generalized gradient-corrected Perdew-Burke-Ernzerhof functional[29], along with a double numerical basis set including p-polarization function (DNP), was applied for all calculations. Dispersion-corrected DFT scheme was used to describe the van der Waals interaction. The solution effect of water was accounted for by using conductor-like screening model. During the coordinates relaxation, the tolerances of energy and force were $1 \times 10^{-5}$ Ha and 0.002 Ha/Å, and the maximum displacement was $5 \times 10^{-3}$ Å, respectively. The Monkhorst-Pack k-point mesh was $4 \times 4 \times 1$ for the graphite plane and $4 \times 1 \times 1$ for other models. The free energies of the reactions at the temperature of 298.15 K were calculated with the partial hessian vibrational analysis method[30]. The free

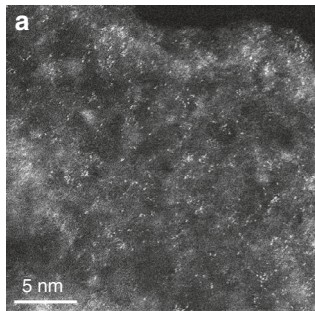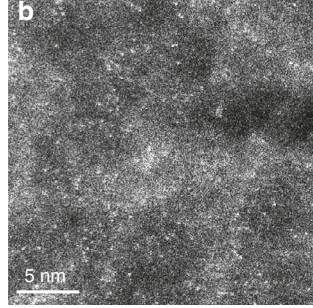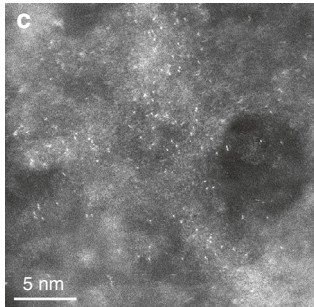

**Fig. 5** HAADF-STEM images. **a** $Pd_1$/hNCNC. **b** $Au_1$/hNCNC. **c** $Ir_1$/hNCNC

energy diagrams were calculated with the method elaborated in our previous publication[31].

## Data availability

All the raw data used in this work are available on request from the authors.

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

## Acknowledgements

This work was jointly supported by the National Key Research and Development Program of China (2017YFA0206500, 2018YFA0209103), National Natural Science Foundation of China (21832003, 21773111, 51571110, 21573107, 11474147, 11874199) and National Basic Research Program of China (2015CB654901). The numerical calculations have been done on the computing facilities in the High Performance Computing Center (HPCC) of Nanjing University. We gratefully thank the XAFS station (BL14W1) of the Shanghai Synchrotron Radiation Facility (SSRF). Z.Q.Z. thanks Mr. Xiao Wang for the UV-vis measurements.

## Author contributions

Z.H. and Q.W. conceived and supervised the project. Z.Q.Z. designed and performed the experiment. Y.G.C. and L.J.Y. carried out the DFT calculations. L.Q.Z., Z.H.A., B.S.Z., and P.W. performed the HAADF-STEM measurements. C.C. and H.Y. helped with the XAFS characterization. L.Y.D. and Y.F.B. helped with the measurements of specific surface area and conductivity. Z.H., Z.Q.Z., Q.W., L.J.Y., X.Z.W, and P.W. evaluated the data and made the intensive discussion. Z.Q.Z., Q.W., and Z.H. wrote the manuscript. The manuscript was revised by all authors.

## Additional information

**Competing interests:** The authors declare no competing interests.

**Journal Peer Review Information**: *Nature Communications* thanks Zhenyu Li, Jingjun Liu, and Ji-Jun Zou for their contribution to the peer review of this work. Peer reviewer reports are available.

