## [Peer Review File · Nature Communications]

Redactions:

Editorial Note: Parts of this peer review file have been redacted as indicated to maintain the confidentiality of unpublished data. When text is deleted in rebuttals and referee reports, add "[redacted]" in that location.

Reviewers' comments:

Reviewer #1 (Remarks to the Author):

In this manuscript, the authors proposed a new facile protocol to prepare single-site catalysts. Single Pt atoms are well distributed in the micropore with the aid of N-anchoring. Extensive experimental characterization has been performed to investigate the local structure and evaluate the catalytic performance. Theoretical calculations have also been performed to provide more insights. Results presented here are interesting, and this manuscript is recommended to be published in Nature Commun. after the following comment is properly responded. The mechanism of SSC formation is proposed to be a synergistic effect from micropore trapping and nitrogen anchoring. The latter is easy to understand but the former is not theoretically well interpreted. For example, why binding for monolayer is not strong as shown in Figure 2? More discussion about micropore trapping is desirable.

Reviewer #2 (Remarks to the Author):

In this manuscript, the authors prepare single-site Pt catalysts on hierarchical nitrogen-doped carbon nanocages via a very simple impregnation-adsorption method, that is a solution adsorption of H_2PtCl_6 anions followed by a mild drying. It is very interesting to explore the adsorption of the anions on by theoretical simulations and experiments. Based on the detail atomic structure of the Pt-support, the HER activity and durability of the catalysts have been discussed. The improved performance is attributed to a synergism of micropore trapping and nitrogen anchoring of carbon support for the metal atoms. It is novel. So, the reviewer recommends this paper for the publication after the following revisions.

1. The authors think that the synthesized Pt-based catalyst exhibits a record-high electrocatalytic hydrogen evolution performance with low overpotential, high mass activity and long stability, much superior to the Pt-based catalysts to date. To confirm this statements, more literature (eg: ACS Catal. 2018, 8, 8450–8458; Nat. Rev. Chem. 2018, 2, 65–81) should be carefully consulted and compared in this regard.

2. Why do the special micropores (~0.6 nm) show the unique advantage to reinforce the interaction with single metal atoms, compared with other pores?

3. The nitrogen content of the home-made hierarchical nitrogen-doped carbon nanocages is up to 9.5 at.%. What's the reason for the so high nitrogen content? For common graphitic N-doped carbons, the nitrogen content is generally about 3 at%.

4. " ... the individual Pt atoms randomly disperse on the hNCNC support with the high percentage of 96.7%" What's the meaning of the data? Is it the loading of Pt on carbon?

5. The Pt/hNCNC has a dominant peak at 1.75 Å. It reveal the formation of the Pt-N/C/O bonds. How to identify them (eg: Pt-N, Pt-O or Pt-C bonds)? What are their contents respectively? If it is Pt-N bond, which type of N atoms coordinated with Pt? Please give detail experimental or literature results.

6. In the dechlorination process, the authors think that the electrons should transfer from Cl atoms to Pt atom. Why is it high for the oxidation state of Pt atoms in Pt1/hNCNC?

The electrostatic interaction between Pt and support plays a key role in deciding the HER electrocatalytic activity over the catalyst. It is possible that the electron transfer from the single Pt

atoms to support or opposite direction. Which one can favors the HER? Please give a reasonable explanation.

7. The free energy of Pt atom bonding with two py-N atoms should be higher than that of Pt bonding with 4 py-N atoms, as suggested by literature.

8. Fig.2, Fig 4(d) and (e) should be re-performed, because they are not clear enough to understand by readers.

9. Line 175: The $[\text{PtCl}_6]^{2-}$ anions are only partially dechlorinated upon heat treatment at 70 °C as confirmed by mass spectroscopy analysis for Pt1/hNCNC sample. How about the residual salt?

10. Line 215: The authors used Ag/AgCl electrode as the reference electrode, and converted all the measured potentials to reversible hydrogen electrode (RHE). The detailed converted method was not given, but this is very important for comparing HER performance of the catalyst with that of others in literature.

11. Line 275: The authors declared "The HER performance is used as an indicator to evaluate the thermal stability of the Pt SSCs." It may be a misleading.

Reviewer #3 (Remarks to the Author):

This manuscript reported a simple and generalized impregnation-adsorption method to construct single-site catalysts (SSCs) of precious metals by taking advantage of the synergism of micropore trapping and nitrogen anchoring for the first time. As known, the SSCs have attracted intensive interests owing to their extraordinary catalytic activity and selectivity, but their facile construction and durable use is a highly challenging yet important topic. By the simplest method reported in this manuscript, the SSCs of Pt, Pd, Au and Ir were facilely obtained. And the Pt SSC presented a record-high activity and stability for HER. The structure-performance correlation was well elucidated by the combination of experimental and theoretical studies, which could provide guidance to the researchers working in this exciting area. The manuscript is also clearly written. Overall, I think this paper is qualified for Nature Communications after addressing the following questions.

1. The $[\text{PtCl}_6]^{2-}$ anions can be adsorbed on the protonated py-N atoms by electrostatic interaction, leading to the formation of individual Pt atoms after dechlorination. It is suggested to discuss whether this impregnation-adsorption method can be used to anchor metal cations.

2. In Figure 2, the two py-N atoms are adjacent in the micropores, but practically the py-N atoms should be randomly distributed. Do the micropores with different distribution of py-N atoms have different adsorption free energies?

3. How about the graphitic N? The XPS measurement shows quite amount of graphitic N existed in the hNCNC. Can they also facilitate the capture of $[\text{PtCl}_6]^{2-}$ anions?

4. The HER stability of Pt1/hNCNC should be compared with that of the Pt SSCs reported in literature.

The point-by-point response to the reviewers' comments

(For clarity, the reviewers' comments are quoted in *italics* ahead of the corresponding responses.)

Reviewer #1

In this manuscript, the authors proposed a new facile protocol to prepare single-site catalysts. Single Pt atoms are well distributed in the micropore with the aid of N-anchoring. Extensive experimental characterization has been performed to investigate the local structure and evaluate the catalytic performance. Theoretical calculations have also been performed to provide more insights. Results presented here are interesting, and this manuscript is recommended to be published in Nature Commun. after the following comment is properly responded. The mechanism of SSC formation is proposed to be a synergic effect from micropore trapping and nitrogen anchoring. The latter is easy to understand but the former is not theoretically well interpreted. For example, why binding for monolayer is not strong as shown in Figure 2? More discussion about micropore trapping is desirable.

General Response:

Thank you very much for the encouraging comments and valuable suggestion.

The micropore trapping mainly comes from the van der Waals interaction between the $[\text{PtCl}_6]^{2-}$ anion and the atoms at the edge (or wall) of micropore. The trapping strength is closely related to the effective atoms involved in the van der Waals interaction. Accordingly, for hCNC, the trapping of $[\text{PtCl}_6]^{2-}$ in the bi-layer micropore is much stronger than the case in the mono-layer micropore, with the corresponding adsorption free energy of 4.3 eV and 2.3 eV. The slightly stronger binding on the graphitic plane (2.8 eV) than in the mono-layer micropore (2.3 eV) is due to the different adsorption configuration. For hNCNC, in addition to the van der Waals interaction, the electrostatic interaction between the protonated pyridinic N atom and $[\text{PtCl}_6]^{2-}$ further strengthens the capability to capture $[\text{PtCl}_6]^{2-}$ anion. (see Figure 2).

This comprehension is also applicable to understand the decreasing trapping effect with increasing the sizes of the micropores. (see Response to Comment 2 of Reviewer 2)

More discussion about the micropore trapping is added in this revision in Page 4:

'The micropore trapping mainly comes from the van der Waals interaction between the $[\text{PtCl}_6]^{2-}$ anion and the atoms at the edge (or wall) of micropore. The trapping strength is closely related to the effective atoms involved in the van der Waals interaction. Accordingly, the trapping of $[\text{PtCl}_6]^{2-}$ in the bi-layer micropore (4.3 eV) is much stronger than the case in the mono-layer micropore (2.3 eV). The matchable size of micropore (~0.6 nm) to $[\text{PtCl}_6]^{2-}$ (~0.5 nm) also contributes a lot to the strong capturing capability.'

Reviewer #2:

In this manuscript, the authors prepare single-site Pt catalysts on hierarchical nitrogen-doped carbon nanocages via a very simple impregnation-adsorption method, that is a solution adsorption of H_2PtCl_6 anions followed by a mild drying. It is very interesting to explore the adsorption of the anions on by theoretical simulations and experiments. Based on the detail atomic structure of the Pt-support, the HER activity and durability of the catalysts have been discussed. The improved performance is attributed to a synergism of micropore trapping and nitrogen anchoring of carbon support for the metal atoms. It is novel. So, the reviewer recommends this paper for the publication after the following revisions.

General Response:

We do appreciate the positive comments from the reviewer, and the point-by-point response has been shown below.

Comment 1: *The authors think that the synthesized Pt-based catalyst exhibits a record-high electrocatalytic hydrogen evolution performance with low overpotential, high mass activity and long stability, much superior to the Pt-based catalysts to date. To confirm this statements, more literature (eg: ACS Catal. 2018, 8, 8450–8458; Nat. Rev. Chem. 2018, 2, 65–81) should be*

carefully consulted and compared in this regard.

Response to Comment 1:

Thank you for the valuable suggestion.

We have carefully checked the HER performances of the related literatures including the two suggested ones. The HER performance of the Pt single atoms in [ACS Catal. 2018, 8, 8450–8458] exhibited an overpotential of 25 mV at 10 mA cm⁻² and a mass activity of 2.86 A mg⁻¹Pt at the overpotential of 25 mV, slightly inferior to ours. The Review paper [Nat. Rev. Chem. 2018, 2, 65–81] compared various non-Pt 1D, 2D and hybrid catalysts, and the HER activity of our Pt₁/hNCNC catalyst still locates at the top-ranking. Hence, our claim on the record-high HER performance of Pt₁/hNCNC catalyst is reliable.

The HER performances in [ACS Catal. 2018, 8, 8450–8458] are incorporated into the revised **Supplementary Table 2**. HER performance of Pt SSCs and Pt-based catalysts in acidic medium.

Comment 2: *Why do the special micropores (~0.6 nm) show the unique advantage to reinforce the interaction with single metal atoms, compared with other pores?*

Response to Comment 2:

The hNCNC and hCNC possess abundant micropores centered at ~0.6 nm (Supplementary Figure 1). Hence, the [PtCl₆]²⁻ anions with the size of ~0.5 nm (slightly < ~0.6 nm) can be trapped by the micropores. Our theoretical calculations indicate, for the adsorption of [PtCl₆]²⁻ in the micropores, the exothermic adsorption free energy decreases with increasing the micropore size [Redacted]. The micropores of ~0.6 nm (6×6 configuration) possess the largest adsorption free energy and the best capturing capability compared with other pores.

Actually, [Redacted] the effective atoms involved in the interaction between the pores and the [PtCl₆]²⁻ anion decrease with increasing pore size. The micropore of ~0.6 nm is only slightly larger than the size of [PtCl₆]²⁻ (~0.5 nm), hence has the most effective atoms, leading to the largest adsorption free energy and the best capturing capability.

We have discussed this point in this revision as follows in page 4:

‘The micropore trapping mainly comes from the van der Waals interaction between the [PtCl₆]²⁻ anion and the atoms at the edge (or wall) of micropore. The trapping strength is closely related to the effective atoms involved in the van der Waals interaction. Accordingly, the trapping of [PtCl₆]²⁻ in the bi-layer micropore (4.3 eV) is much stronger than the case in the mono-layer micropore (2.3 eV). The matchable size of micropore (~0.6 nm) to [PtCl₆]²⁻ (~0.5 nm) also contributes a lot to the strong capturing capability.’

[Redacted]

Comment 3: *The nitrogen content of the home-made hierarchical nitrogen-doped carbon nanocages is up to 9.5 at.%. What’s the reason for the so high nitrogen content? For common graphitic N-doped carbons, the nitrogen content is generally about 3 at%.*

Response to Comment 3:

Different from the common graphitic N-doped carbons, the hNCNC was prepared by the in situ MgO template method via the pyrolysis of pyridine with high N content, which was developed by our group [Adv. Mater. 2012, 24, 5593]. The resultant carbon coating layer possesses abundant defects and plentiful micropores, and the N atoms are easy to incorporate into the carbon matrix, leading to the high N content. Actually, the N content of hNCNC decreases with increasing the preparation temperature (T). The N content is above 9 at% when T is below 800 °C [see: Adv. Mater. 2012, 24, 5593]. The hNCNC used here is prepared at 800 °C with N content of 9.5 at.%, in agreement with our previous report.

Comment 4: *“ ... the individual Pt atoms randomly disperse on the hNCNC support with the high percentage of 96.7%” What’s the meaning of the data? Is it the loading of Pt on carbon?*

Response to Comment 4:

The datum “96.7%” means the ratio of Pt single atoms to all the Pt dots (including Pt single atoms and clusters), which is a statistic datum from the HAADF-STEM observation. For clarity, this sentence is revised as: ...the individual Pt atoms randomly dispersed on the hNCNC support, which account for a high percentage of 96.7% in all the Pt dots.

Comment 5: *The Pt/hNCNC has a dominant peak at 1.75 Å. It reveal the formation of the Pt-N/C/O bonds. How to identify them (eg: Pt-N, Pt-O or Pt-C bonds)? What are their contents respectively? If it is Pt-N bond, which type of N atoms coordinated with Pt? Please give detail experimental or literature results.*

Response to Comment 5:

The Pt-N/C/O bonds are hardly deconvolved from the peak at 1.75 Å because they give similar scattering parameters due to their neighboring positions in the periodic table [Yi Xie, et al., *Adv. Mater.* 2016, 28, 2427]. Similar to the recent literatures [Weilin Xu, et al., *Nat. Commun.* 2017, 8, 15938; Yi Xie, et al., *Adv. Mater.* 2016, 28, 2427], the dominant peaks at 1.75 Å are ascribed to Pt-N/C/O bonds in this paper.

The type of N atoms coordinated with Pt should be pyridinic N owing to the stronger bonding to Pt atoms for pyridinic N than graphitic N, as learnt from our previous theoretical study [Zheng Hu, et al., *J. Mater. Chem.*, 2010, 20, 1702]. Specifically, the pyridinic N atoms at the edges of the micropores are protonated and positively charged in the H₂PtCl₆ solution, leading to the formation of stable ion pair of [C_x(NH)₂]²⁺[PtCl₆]²⁻ via the electrostatic interaction. This statement is provided in Page 5 of the manuscript.

Comment 6: *In the dechlorination process, the authors think that the electrons should transfer from Cl atoms to Pt atom. Why is it high for the oxidation state of Pt atoms in Pt₁/hNCNC? The electrostatic interaction between Pt and support plays a key role in deciding the HER electrocatalytic activity over the catalyst. It is possible that the electron transfer from the single Pt atoms to support or opposite direction. Which one can favors the HER? Please give a reasonable explanation.*

Response to Comment 6:

In the dechlorination process, the Pt⁴⁺ species are reduced to Pt single atoms resting on the carbon-based supports. Owing to the Pt-support interaction, the electron transfers from the Pt atoms to the supports, leading to the high oxidation state of Pt single atoms relative to Pt foil and nanoparticles, in agreement with the cases in literatures [Tao Zhang, et al., *Nat. Chem.* 2011, 3, 634; Weilin Xu, et al., *Nat. Commun.* 2017, 8, 15938].

The electron transfer from the Pt single atoms to the support favors the HER. As revealed in literatures [Xueliang Sun, et al., *Nat. Commun.* 2016, 7, 13638; Tongbu Lu, et al., *Angew. Chem. Int. Ed.* 2018, 57, 9382], the charge transfer from Pt to the support would cause more unoccupied 5d orbitals of Pt single atoms, which can interact strongly with the 1s orbital of the H atoms, leading to the electron pairing and hydride formation, therefor the high HER activity. Our experimental results also demonstrate the high HER activity of Pt₁/hNCNC (Figure 3).

We have discussed this point in this revision as follows in page 6:

‘The excellent HER performance should result from the charge transfer from Pt to the support as reflected by the high oxidation state of Pt atoms in Pt₁/hNCNC (Fig. 1c), which leads to the unoccupied 5d orbitals of Pt single atoms and thus favors the HER²⁰,

Comment 7: *The free energy of Pt atom bonding with two py-N atoms should be higher than that of Pt bonding with 4 py-N atoms, as suggested by literature.*

Response to Comment 7:

Yes, we agree that the Pt atoms bonding with 4 py-N atoms are more stable than that bonding with two py-N atoms. However, in our case, the Pt single atoms are generated by the simplest

impregnation-adsorption and the subsequent drying process at ~70 °C. Such a mild synthetic condition will not induce the structural rearrangement of the supports, hence will not form the Pt atoms bonding with 4 py-N atoms.

Comment 8: Fig.2, Fig 4(d) and (e) should be re-performed, because they are not clear enough to understand by readers.

Response to Comment 8:

Thanks. We have re-performed Fig. 2, Fig. 4(d) and (e) in the revised manuscript.

Comment 9: Line 175: The $[\text{PtCl}_6]^{2-}$ anions are only partially dechlorinated upon heat treatment at 70 °C as confirmed by mass spectroscopy analysis for Pt₁/hNCNC sample. How about the residual salt?

Response to Comment 9:

The Cl-related signals in the MS analysis of Pt₁/hNCNC should be attributed to the Cl-containing species which were formed during the dechlorination of $[\text{PtCl}_6]^{2-}$ in the synthesis and partially adsorbed on hNCNC support, rather than from the residual $[\text{PtCl}_6]^{2-}$. For supporting this, hNCNC was immersed in the HCl solution (denoted as hNCNC-HCl), followed by filtrating, washing with distilled water and ethanol repeatedly, and drying at 70 °C. And then, the so-obtained hNCNC-HCl was detected by MS analysis upon heating.

As shown in Fig. R1, the Cl species such as CCl, CCl₂ and CCl₃ were detected for hNCNC-HCl. This result indicates that the Cl-containing species can be adsorbed/bonded with carbon support and then be desorbed/decomposed upon heating. In the synthesis of Pt₁/hNCNC, the Cl species dissociated from the $[\text{PtCl}_6]^{2-}$ could be adsorbed or bonded on the hNCNC, leading to the signals in the MS analysis. Actually, the dechlorination starts to occur at ca. 50 °C (below 70 °C) even at the programmed temperature condition (see the Pt/hNCNC-freeze sample in Supplementary Figure 9). Our synthesis took a long time of 10 h under 70 °C, which should lead to the negligible residual $[\text{PtCl}_6]^{2-}$.

In addition, the EXAFS peak corresponding to the Pt-Cl bonds (~2.0 Å) is absent in the Pt₁/hNCNC sample (Fig. 1d), further indicating the negligible Pt-Cl species in this catalyst [Yi Xie, et al., *Adv. Mater.* 2016, 28, 2427; Ning Yan, et al., *Nat. Commun.* 2017, 8, 16100; Tongbu Lu, et al., *Angew. Chem. Int. Ed.* 2018, 57, 9382].

This supplemented experiment has been added into the revised version as Supplementary Figure 9b.

Fig. R1. Programmed-temperature mass spectrometry for hNCNC-HCl at a heating rate of $10\text{ }^{\circ}\text{C min}^{-1}$ from 25 to $450\text{ }^{\circ}\text{C}$. Note: m/z (mass-to-charge ratio) of 47 and 49 correspond to CCl ; 82, 84 and 86 to CCl_2 ; 117 and 119 to CCl_3 .

Comment 10: Line 215: The authors used Ag/AgCl electrode as the reference electrode, and converted all the measured potentials to reversible hydrogen electrode (RHE). The detailed converted method was not given, but this is very important for comparing HER performance of the catalyst with that of others in literature.

Response to Comment 10:

Thanks for your suggestion. We have added the converted equation, $E_{\text{RHE}} = E_{\text{Ag}/\text{AgCl}} + 0.209 + 0.059 \times \text{pH}$, in “Methods” part in the revised manuscript.

Comment 11: Line 275: The authors declared “The HER performance is used as an indicator to evaluate the thermal stability of the Pt SSCs.” It may be a misleading.

Response to Comment 11:

Thanks for your reminding. This sentence is deleted in the revised version.

Reviewer #3:

This manuscript reported a simple and generalized impregnation-adsorption method to construct single-site catalysts (SSCs) of precious metals by taking advantage of the synergism of micropore trapping and nitrogen anchoring for the first time. As known, the SSCs have attracted intensive interests owing to their extraordinary catalytic activity and selectivity, but their facile construction and durable use is a highly challenging yet important topic. By the simplest method reported in this manuscript, the SSCs of Pt, Pd, Au and Ir were facilely obtained. And the Pt SSC presented a record-high activity and stability for HER. The structure-performance correlation was well elucidated by the combination of experimental and theoretical studies, which could provide guidance to the researchers working in this exciting area. The manuscript is also clearly written. Overall, I think this paper is qualified for Nature Communications after addressing the following questions.

General Response:

Thank you for the encouraging comments.

Comment 1: The $[\text{PtCl}_6]^{2-}$ anions can be adsorbed on the protonated py-N atoms by electrostatic interaction, leading to the formation of individual Pt atoms after dechlorination. It is suggested to discuss whether this impregnation-adsorption method can be used to anchor metal cations.

Response to Comment 1:

In neutral or weakly basic solutions, the metal cations with empty orbitals could be anchored on the py-N atoms with lone pair electrons owing to the coordination effect. Actually, this is what we are trying to do, i.e., constructing the SSCs of non-precious metals by this strategy.

Comment 2: In Figure 2, the two py-N atoms are adjacent in the micropores, but practically the py-N atoms should be randomly distributed. Do the micropores with different distribution of py-N atoms have different adsorption free energies?

Response to Comment 2:

Actually, we have performed a series of calculations based on the bi-layer micropore models with two py-N atoms positioning at different sites, as shown in Fig. R2. The adsorption free energies for these configurations are all at the same level with a small difference in the range of 4.8~4.9 eV. This indicates that the different distribution of py-N atoms on the edge of micropores has little influence on the adsorption free energy. Thus we use the typical one, i.e. 1 in Fig. R2 (or 6 in Fig. 2), to represent the case of graphitic bi-layer with the micropore decorated by two py-N atoms.

Fig. R2. Different configurations of $[\text{PtCl}_6]^{2-}$ in the bi-layer micropores (~ 0.6 nm) decorated by two py-N atoms at different positions and corresponding adsorption free energies. 1,2,3, Two py-N atoms with different distances on the same layer. 4, Vertically aligned two py-N atoms on different layers. 5, Staggered two py-N atoms on different layers.

Comment 3: How about the graphitic N? The XPS measurement shows quite amount of graphitic N existed in the hNCNC. Can they also facilitate the capture of $[\text{PtCl}_6]^{2-}$ anions?

Response to Comment 3:

We calculated the adsorption free energies of $[\text{PtCl}_6]^{2-}$ anion on the graphene sheet with graphitic N as shown in Fig. R3. The graphitic N dopant has little contribution to the adsorption free energy, which is quite similar to the graphene sheet.

We have added the Fig. R3 into the revised version as Supplementary Figure 8.

Fig. R3. The configurations and corresponding adsorption free energies of [PtCl₆]²⁻ on (1) graphene sheet, (2) graphene sheet decorated by one graphitic N atom, (3) graphene sheet decorated by two graphitic N atoms.

Comment 4: The HER stability of Pt₁/hNCNC should be compared with that of the Pt SSCs reported in literature.

Response to Comment 4:

We have summarized the typical HER stability of the Pt SSCs in literatures as shown in Table R1, which indicates the top-level HER stability of Pt₁/hNCNC.

Table R1. HER stability of Pt SSCs in acidic medium.

Sample	Scan rate (mV s ⁻¹)	Potential range (V vs RHE)	Cycles (times)	Ref.
Pt ₁ /hNCNC-2.92	100	0.23~-0.17	10000	This study
Mo ₂ TiC ₂ T _X -Pt _{SA} ^(a)	20	N.A.	10000	Nat. Catal. 2018, 1, 985.
Pt ₁ /mesoporous carbon	100	0.3~-0.1	1000	Nat. Commun. 2017, 8, 1490
Pt-graphdiyne	5	N.A.	1000	Angew. Chem. Int. Ed. 2018, 57, 9382
Pt ₁ /N-doped porous carbon	100	0.65~1.05	3000	ACS Catal. 2018, 8, 8450-8458
Pt ₁ /MoO _{3-x} /C	50	not available	1000	ChemCatChem 2018, 10, 946

(a): Mo₂TiC₂T_X-Pt_{SA}: double transition metal MXene-Pt single atoms.

REVIEWERS' COMMENTS:

Reviewer #1 (Remarks to the Author):

The revised manuscript is recommended to be published in Nature Communications.

Reviewer #2 (Remarks to the Author):

I think that my concerns have been addressed. So, this manuscript is recommended to be published in Nature Commun.

Reviewer #3 (Remarks to the Author):

The manuscript is now acceptable for publish.